# Psychological consequences of COVID-19 home confinement: The ECLB-COVID19 multicenter study

Achraf Ammar[1,2]*, Patrick Mueller[3,4], Khaled Trabelsi[5,6], Hamdi Chtourou[5,7], Omar Boukhris[5,7], Liwa Masmoudi[5], Bassem Bouaziz[8], Michael Brach[9], Marlen Schmicker[3], Ellen Bentlage[9], Daniella How[9], Mona Ahmed[9], Asma Aloui[7,10], Omar Hammouda[5,11], Laisa Liane Paineiras-Domingos[12,13], Annemarie Braakman-jansen[14], Christian Wrede[14], Sophia Bastoni[14,15], Carlos Soares Pernambuco[16], Leonardo Jose Mataruna-Dos-Santos[17], Morteza Taheri[18], Khadijeh Irandoust[18], Aïmen Khacharem[19], Nicola L. Bragazzi[20,21], Jad Adrian Washif[22], Jordan M. Glenn[23], Nicholas T. Bott[24], Faiez Gargouri[8], Lotfi Chaari[25], Hadj Batatia[25], Samira C. khoshnami[26], Evangelia Samara[27], Vasiliki Zisi[28], Parasanth Sankar[29], Waseem N. Ahmed[30], Gamal Mohamed Ali[31], Osama Abdelkarim[31,32], Mohamed Jarraya[5], Kais El Abed[5], Mohamed Romdhani[7], Nizar Souissi[7], Lisette Van Gemert-Pijnen[14], Stephen J. Bailey[33], Wassim Moalla[5], Jonathan Gómez-Raja[34], Monique Epstein[35], Robbert Sanderman[36], Sebastian Schulz[37], Achim Jerg[37], Ramzi Al-Horani[38], Taysir Mansi[39], Mohamed Jmail[40], Fernando Barbosa[41], Fernando Ferreira-Santos[41], Boštjan Šimunič[42], Rado Pišot[42], Andrea Gaggioli[43,44], Piotr Zmijewski[45], Jürgen M. Steinacker[37], Jana Strahler[46], Laurel Riemann[47], Bryan L. Riemann[48], Notger Mueller[3,4], Karim Chamari[49,50©], Tarak Driss[2©], Anita Hoekelmann[1©], for the ECLB-COVID19 Consortium¶

1 Institute of Sport Science, Otto-von-Guericke University, Magdeburg, Germany, 2 Interdisciplinary Laboratory in Neurosciences, Physiology and Psychology: Physical Activity, Health and Learning, UFR STAPS, UPL, Paris Nanterre University, Nanterre, France, 3 Research Group Neuroprotection, German Center for Neurodegenerative Diseases (DZNE), Magdeburg, Germany, 4 Medical Faculty, Department of Neurology, Otto-von-Guericke University, Magdeburg, Germany, 5 High Institute of Sport and Physical Education of Sfax, University of Sfax, Sfax, Tunisia, 6 Research Laboratory: Education, Motricity, Sport and Health, EM2S, LR19JS01, University of Sfax, Sfax, Tunisia, 7 Physical Activity, Sport, and Health, UR18JS01, National Observatory of Sport, Tunis, Tunisia, 8 Higher Institute of Computer Science and Multimedia of Sfax, University of Sfax, Sfax, Tunisia, 9 Institute of Sport and Exercise Sciences, University of Münster, Münster, Germany, 10 High Institute of Sport and Physical Education, University of Gafsa, Gafsa, Tunisia, 11 Research Laboratory, Molecular Bases of Human Pathology, LR12ES17, Faculty of Medicine, University of Sfax, Sfax, Tunisia, 12 Universidade do Estado do Rio de Janeiro, Rio de Janeiro, Brazil, 13 Faculdade Bezerra de Araújo, Rio de Janeiro, Brazil, 14 Department of Psychology, Health & Technology, University of Twente, Enschede, The Netherlands, 15 Department of Psychology, Università Cattolica del Sacro Cuore, Milano, Italy, 16 Laboratório de Fisiologia do Exercício, Estácio de Sá University, Rio de Janeiro, Brasil, 17 Faculty of Management, Sport Management Department, Canadian University of Dubai, Dubai, United Arab Emirates, 18 Faculty of Social Science, Imam Khomeini International University, Qazvin, Iran, 19 UVHC, DeVisu, Valenciennes; LIRTES-EA 7313. Université Paris Est Créteil Val de Marne, Créteil, France, 20 Department of Health Sciences, Postgraduate School of Public Health, University of Genoa, Genoa, Italy, 21 Department of Mathematics and Statistics, Laboratory for Industrial and Applied Mathematics, York University, Toronto, ON, Canada, 22 Sports Performance Division, National Sports Institute of Malaysia, Kuala Lumpur, Malaysia, 23 Exercise Science Research Center, Department of Health, Human Performance and Recreation, University of Arkansas, Fayetteville, Arkansas, United States of America, 24 Clinical Excellence Research Center, Department of Medicine, Stanford University School of Medicine, Stanford, California, United States of America, 25 Computer Science Department, University of Toulouse, IRIT-INP-ENSEEIHT, Toulouse, France, 26 UFR STAPS, UPL, Paris Nanterre University, Nanterre, France, 27 Onassis Cardiac Surgery Center, Athens, Greece, 28 Department of Physical Education and Sports Sciences, University of Thessaly, Volos, Greece, 29 Consultant in Internal Medicine and Diabetes, MGM Muthoot Hospitals Pathanamthitta, Pathanamthitta, Kerala, India, 30 Consultant Family Physician, CRAFT Hospital and Research Centre, Kodungallur, Kerala, India, 31 Faculty of Physical Education, Assiut University, Assiut, Egypt, 32 Institute for Sports and Sports Science, Karlsruher Institut für Technologie, Karlsruher, Germany, 33 School of Sport, Exercise and Health Sciences, Loughborough University, Loughborough, United Kingdom, 34 FundeSalud, Dept. of Health and Social Services,

**Data Availability Statement:** The data set of the present manuscript include information related to emotional status of human participants collected via electronic survey. In the consent of participation

survey participants were assured all data would be used only for research purposes and data set will not be available for public as advised by the Otto von Guericke University Ethics Committee, Magdeburg, Germany. Therefore, data are available from the corresponding author (ammar1.achraf@ovgu.de) as well as from the "Data Protection Officer" of Otto von Guericke University (datenschutz@ovgu.de) on reasonable request related to research purpose such as Validation, replication, reanalysis, new analysis, reinterpretation or inclusion into meta-analyses.

**Funding:** The authors received no specific funding for this work. PharmIAD, Inc, Savannah, GA, USA provided support in the form of salaries for Laurel Riemann, but did not have any additional role in the study design, data collection and analysis, decision to publish, or preparation of the manuscript. The specific roles of this author are articulated in the 'author contributions' section.

**Competing interests:** All authors have completed the Unified Competing Interest form (available on request from the corresponding author). Authors except Laura Riemann declare: no support from any organisation for the submitted work; no financial relationships with any organisations that might have an interest in the submitted work in the previous three years, no other relationships or activities that could appear to have influenced the submitted work. Laura Riemann declare to have a commercial affiliation "PharmIAD, Inc, Savannah, GA, USA". This funder provided support in the form of salaries for Laurel Riemann, but did not have any additional role in the study design, data collection and analysis, decision to publish, or preparation of the manuscript. The specific roles of this author are articulated in the 'author contributions' section. This commercial affiliation does not alter our adherence to PLOS ONE policies on sharing data and materials.

Government of Extremadura, Merida, Spain, **35** The E-Senior Association, Paris, France, **36** Department of Health Psychology, University Medical Center Groningen, University of Groningen, Groningen, The Netherlands, **37** Sports- and Rehabilitation Medicine, Ulm University Hospital, Ulm, Germany, **38** Department of Exercise Science, Yarmouk University, Irbid, Jordan, **39** Faculty of Physical Education, The University of Jordan, Amman, Jordan, **40** Digital Research Centre of Sfax, Sfax, Tunisia, **41** Laboratory of Neuropsychophysiology, Faculty of Psychology and Education Sciences, University of Porto, Porto, Portugal, **42** Institute for Kinesiology Research, Science and Research Centre Koper, Koper, Slovenia, **43** Department of Psychology, Università Cattolica del Sacro Cuore, Milan, Italy, **44** I.R.C.C.S. Istituto Auxologico Italiano, Milan, Italy, **45** Jozef Pilsudski University of Physical Education in Warsaw, Warsaw, Poland, **46** Department of Psychology and Sport Science, University of Gießen, Gießen, Germany, **47** PharmD, BCBS; PharmIAD, Inc, Savannah, GA, United States of America, **48** Department of Health Sciences and Kinesiology, Georgia Southern University, Statesboro, GA, United States of America, **49** ASPETAR, Qatar Orthopaedic and Sports Medicine Hospital, Doha, Qatar, **50** Laboratory "Sport Performance Optimization", ISSEP Ksar-Said, Manouba University, Manouba, Tunisia

☯ These authors contributed equally to this work.
¶ Membership of the ECLB-COVID19 Consortium is listed in the Acknowledgments.
* ammar1.achraf@ovgu.de

# Abstract

## Background

Public health recommendations and government measures during the COVID-19 pandemic have enforced restrictions on daily-living. While these measures are imperative to abate the spreading of COVID-19, the impact of these restrictions on mental health and emotional wellbeing is undefined. Therefore, an international online survey (ECLB-COVID19) was launched on April 6, 2020 in seven languages to elucidate the impact of COVID-19 restrictions on mental health and emotional wellbeing.

## Methods

The ECLB-COVID19 electronic survey was designed by a steering group of multidisciplinary scientists, following a structured review of the literature. The survey was uploaded and shared on the Google online-survey-platform and was promoted by thirty-five research organizations from Europe, North-Africa, Western-Asia and the Americas. All participants were asked for their mental wellbeing (SWEMWS) and depressive symptoms (SMFQ) with regard to "during" and "before" home confinement.

## Results

Analysis was conducted on the first 1047 replies (54% women) from Asia (36%), Africa (40%), Europe (21%) and other (3%). The COVID-19 home confinement had a negative effect on both mental-wellbeing and on mood and feelings. Specifically, a significant decrease ($p < .001$ and **Δ% =** 9.4%) in total score of the SWEMWS questionnaire was noted. More individuals (+12.89%) reported a low mental wellbeing "during" compared to "before" home confinement. Furthermore, results from the mood and feelings questionnaire showed a significant increase by 44.9% ($p < .001$) in SMFQ total score with more people (+10%) showing depressive symptoms "during" compared to "before" home confinement.

## Conclusion

The ECLB-COVID19 survey revealed an increased psychosocial strain triggered by the home confinement. To mitigate this high risk of mental disorders and to foster an Active and Healthy Confinement Lifestyle (AHCL), a crisis-oriented interdisciplinary intervention is urgently needed.

## Introduction

An unexplained severe respiratory infection detected in Wuhan City of Hubei Province of China was reported to the World Health Organization (WHO) office in China on December 31, 2019. The WHO announced that the disease is caused by a new coronavirus, called COVID-19, which is the acronym of "coronavirus disease 2019" [1]. This new virus has quickly spread worldwide. As of 14 April 2020, a total of 1.910.507 confirmed cases globally with 123.348 deaths had been reported by WHO [2]. Considering the challenges imposed by the COVID-19 pandemic to health care systems and society in general, and in order to cut the rate of new infections and flatten the COVID-19 contagion curve, the majority of countries worldwide imposed mass home-confinement directives, with most including quarantine and physical distancing [3, 4]. Quarantine, and the resulting social isolation, can be major stressors that can contribute to widespread emotional distress [5–8], and may aggravate pre-existing disease [9] and cause disease such as sleep disorder or a weakened immune system [10].

Mental health is an essential component of public health and is associated with a reduced risk of several chronic diseases (e.g. dementia, depression, obesity, coronary heart disease), premature morbidity, and functional decline [11, 12]. According to the WHO, mental health is "a state of wellbeing in which the individual realizes his or her own abilities, can cope with the normal stresses of life, can work productively and fruitfully, and is able to make a contribution to his or her community" [13]. There are many important facets to mental health such as personal freedoms, financial security, social stability and individual lifestyle factors (e.g. physical activity).

Unfortunately, many of the social and individual consequences of the COVID-19 pandemic impose upon these facets. For example, the uncertainty of prognosis, seclusion as a result of quarantine, and financial losses associated with a reduction in economic activity likely result in several severe emotional reactions (e.g., distress) and unhealthy behaviors (e.g. excessive substance use). In this context, a recent review by Brooks et al. [14] reported negative psychological effects, including depression, stress, fear, confusion, and anger, in quarantined people during previous epidemic. Specifically, infringement upon personal freedoms, duration of confinement, resulting financial losses, and insufficient medical care have all been suggested to increase risk for psychiatric illness during quarantine [5]. This notion, the negative effects of quarantine on mental health including psychological and emotional problems (e.g., depression and anxiety), is directly supported by earlier studies during several outbreaks of previous infections (e.g., SARS) [15, 16].

In contrast to the above earlier investigation of relatively recent infections, the dimension of the current COVID-19 pandemic drastically exceeds the previous quarantine measures, as well as the financial hardships, on an international scale. In this regard, there resides the chance of a secondary public mental health sequela related to the impact of COVID-19 that extends beyond the immediate physical health crises suggesting the need to investigate the effects of COVID-19 home confinement on mental health in detail. Therefore, an international

online survey (ECLB-COVID19) was launched in April 6, 2020 in multiple languages to elucidate the emotional consequences of COVID-19 home confinement. This study is the first translational large-scale survey on mental health and emotional wellbeing in the general population during the COVID-19 pandemic. It can be assumed that the COVID-19 pandemic will have negative implications for individual and collective mental health.

The present paper presents data on mental wellbeing, mood and feeling before and during home confinement. Other parts of the survey evaluate physical activity and diet behaviors [7], social participation and life satisfaction [17] and mental health and general lifestyle [18, 19]; these findings are published elsewhere. All papers share a common method description.

## Materials and methods

We report findings on the first 1047 replies to an international online-survey on mental health and multi-dimension lifestyle behaviors during home confinement (ECLB-COVID19). ECLB-COVID19 was opened on April 1, 2020, tested by the project's steering group for a period of 1 week, before starting to spread it worldwide on April 6, 2020 [6, 7, 17, 18]. Thirty-five research organizations from Europe, North-Africa, Western Asia and the Americas promoted dissemination and administration of the survey. ECLB-COVID19 was administered in English, German, French, Arabic, Spanish, Portuguese, and Slovenian languages. The survey included sixty-four questions on health, mental wellbeing, mood, life satisfaction and multidimension lifestyle behaviors (i.e., physical activity, diet, social participation, sleep, technology-use, need of psychosocial support). All questions were presented in a differential format, to be answered directly in sequence with regard to both "before" and "during" confinement conditions [6, 7, 17, 18]. The study was conducted according to the Declaration of Helsinki. The protocol and the consent form were fully approved (identification code: 62/20) by the Otto von Guericke University Ethics Committee, Magdeburg, Germany.

### Survey development and promotion

The cross-sectional ECLB-COVID19 electronic survey was designed by a steering group of multidisciplinary scientists and academics (i.e., human science, sport science, neuropsychology and computer science) at the University of Magdeburg (principal investigator), the University of Sfax, the University of Münster and the University of Paris-Nanterre, following a structured review of the literature. The survey was then reviewed and edited by 50 colleagues and experts worldwide. The survey was uploaded and shared on the Google online survey platform. A link to the electronic survey was distributed worldwide by consortium colleagues via a range of methods: invitation via e-mails, shared in consortium's faculties official pages, ResearchGate™, LinkedIn™ and other social media platforms such as Facebook™, WhatsApp™ and Twitter™. Public were also involved in the dissemination plans of our research through the promotion of the ECLB-COVID19 survey in their networks. The survey included an introductory page describing the background and the aims of the survey, the consortium, ethics information for participants and the option to choose one of seven available languages (English, German, French, Arabic, Spanish, Portuguese, and Slovenian). The present study focuses on the first thousand responses (i.e., 1047 participants), which were reached on April 11, 2020, approximately one-week after the survey began. This survey was open for all people worldwide aged 18 years or older. People with cognitive decline are excluded [6, 7, 17–19].

### Data privacy and consent of participation

During the informed consent process, survey participants were assured all data would be used only for research purposes and data set will not be available for public. Participants' answers

were anonymous and confidential according to Google's privacy policy [7, 17–19]. Participants did not have to mention their names or contact information. In addition, participants could stop participating in the study and could leave the questionnaire at any stage before the submission process and their responses were not saved. Response were saved only by clicking on "submit" button. By completing the survey, participants were acknowledging the above approval form and were consenting to voluntarily participate in this anonymous study. Participants have been requested to be honest in their responses.

## Survey questionnaires

The ECLB-COVID19 is a translational electronic survey designed to assess emotional and behavioral change associated with home confinement during the COVID-19 outbreak. Therefore, a collection of validated and/or crisis-oriented brief questionnaires were included (Ammar et al. 2020a-e). These questionnaires assess mental wellbeing (Short Warwick-Edinburgh Mental Wellbeing Scale (SWEMWBS)) [18–20], mood and feeling (Short Mood and Feelings Questionnaire (SMFQ)) [18, 19, 21], life satisfaction (Short Life Satisfaction Questionnaire for Lockdowns (SLSQL)) [17, 19], social participation (Short Social Participation Questionnaire for Lockdowns (SSPQL) [17, 19), physical activity (International Physical Activity Questionnaire Short Form (IPAQ-SF)) [6, 7, 19, 22], diet behaviours (Short Diet behaviours Questionnaire for Lockdowns (SDBQL)) [6, 7, 19], sleep quality (Pittsburgh Sleep Quality Index (PSQI)) [23], and some key questions assessing the technology-use behaviours (Short Technology-use Behaviours Questionnaire for Lockdowns (STBQL)), demographic information, and the need of psychosocial support [19]. Reliability of the shortened and/or newly adopted questionnaires was tested by the project steering group through piloting, prior to survey administration. These brief crisis-oriented questionnaires demonstrated high to excellent test-retest reliability coefficients (r = 0.84–0.96). A multi-language validated version already existed for the majority of these questionnaires and/or questions. However, for questionnaires that did not already exist in multi-language versions, we followed the procedure of translation and back-translation, with an additional review for all language versions from the international scientists of our consortium. In this manuscript, we report only results on mental wellbeing (SWEMWBS), mood, and feeling (SMFQ). A copy of the complete survey can be found in S1 File.

**The Short Warwick-Edinburgh Mental Wellbeing Scale (SWEMWBS).** The SWEMWBS is a short version of the Warwick–Edinburgh Mental Wellbeing Scale (WEMWBS). The WEMWBS was developed to enable the monitoring of mental wellbeing in the general population and in response to projects, programmes and policies focusing on mental wellbeing. The SWEMWBS uses seven of the WEMWBS's 14 statements about thoughts and feelings, which relate more to functioning than feelings suggesting an ability to detect clinically meaningful change [24, 25]. The seven statements are positively worded with five response categories from 'none of the time (score 1)' to 'all of the time (score 5)'. The SWEMWBS was recently validated for the general population and is scored by first summing the scores for each of the seven items, which are scored from 1 to 5 [20]. The total raw scores are then transformed into metric scores using the SWEMWBS conversion table. Total scores range from 7 to 35 with higher scores indicating higher positive mental wellbeing. Based on scores that were at least one standard deviation below and above the mean, respectively [26], categories for SWEMWBS were considered 'low' (7–19.3), 'medium' (20.0–27.0) and 'high' (28.1–35) mental wellbeing [20].

**The Short Mood and Feelings Questionnaire (SMFQ).** The SMFQ is a short version of the Mood and Feelings Questionnaire (MFQ) developed by Costello and Angold [27].

The SMFQ was developed in response to the need for a brief depression measure [28]. The SMFQ is, therefore, suggested as a brief screening tool for depression based on thirteen of the MFQ's 33 statements about how the subject has been feeling or acting recently [21]. The MFQ is scored by summing together the point values of responses for each item ("not true" = 0 points; "sometimes true" = 1 point; "true" = 2 points) with higher scores on the SMFQ suggesting more severe depressive symptoms. Scores on SMFQ range from 0 to 26. A total score of 12 or higher may indicate the presence of depression in the respondent [18, 21].

## Data analysis

Descriptive statistics were used to define the proportion of responses for each question and the distribution of the total score of both questionnaires. All statistical analyses were performed using the commercial statistical software STATISTICA (StatSoft, Paris, France, version 10.0) and Microsoft Excel 2010. Normality of the data distribution in each question was confirmed using the Shapiro-Wilks-W-test. Values were computed and reported as mean ± SD (standard deviation). To assess for significant differences in responses with reference to "before" and "during" the confinement period, paired samples t-tests were used for normally distributed data (responses to the *SWEMWBS* questionnaire) and the Wilcoxon test was used when normality was not assumed (responses to the SMFQ). Effect size (Cohen's d) was calculated to determine the magnitude of the change of the score and was interpreted using the following criteria: $0.2 \leq d < 0.5$: small, $0.5 \leq d < 0.8$: moderate, and $d \geq 0.8$: large [29]. Statistical significance was set at $\alpha < 0.05$.

## Results

### Sample description

The present study focused on the first thousand responses (i.e., 1047 participants). Overall, 54% of the participants were women, and the participants were from Western Asia (36%), North Africa (40%), Europe (21%) and other (3%). Age, health status, employment status, level of education and marital status are presented in Table 1.

### The Short Warwick-Edinburgh Mental Wellbeing Scale (SWEMWBS)

Change in mental wellbeing score assessed through the SWEMWBS from "before" to "during" confinement period are presented in Table 2. The total score decreased significantly by 9.4% during compared to before home confinement ($t = 18.82$, $p < .001$, $d = 0.58$). A statistically significant decrease was observed for each of the 7 questions. Particularly, feeling related questions such as feeling optimistic, useful, relaxed and close to others showed a lower score at "during" compared to "before" confinement with |Δ%| ranged from 4% to 13% ($3.44 \leq t \leq 20.26$; $p < .001$; $0.106 \leq d \leq 0.626$). Similarly, participants scored lower in thinking related questions "during" compared to "before" confinement period with |Δ%| ranged from 7% to 16% for the capacities to deal well with problems, think clearly and make up own mind about things ($10.36 \leq t \leq 12.89$, $p < .001$, $0.32 \leq d \leq 0.51$). For detailed distribution of responses (in %) please see S1 Table.

### The Short Mood and Feelings Questionnaire (SMFQ)

Change in mood and feeling score from "before" to "during" confinement period in response to SMFQ depression monitoring tool are presented in Table 3. The SMFQ total score increased significantly by 44.9% "during" compared to "before" home confinement ($z = 14.52$, $p < .001$,

**Table 1. Demographic characteristics of the participants.**

| Variables | | N | (%) |
|---|---|---|---|
| **Gender** | | | |
| | Male | 484 | (46.2%) |
| | Female | 563 | (53.8%) |
| **Continent** | | | |
| | North Africa | 419 | (40%) |
| | Western Asia | 377 | (36%) |
| | Europe | 220 | (21%) |
| | Other | 31 | (3%) |
| **Age (years)** | | | |
| | 18–35 | 577 | (55.1%) |
| | 36–55 | 367 | (35.1%) |
| | >55 | 103 | (9.8%) |
| **Level of Education** | | | |
| | Master/doctorate degree | 527 | (50.3%) |
| | Bachelor's degree | 397 | (37.9%) |
| | Professional degree | 28 | (2.7%) |
| | High school graduate, diploma or the equivalent | 69 | (6.6%) |
| | No schooling completed | 26 | (2.5%) |
| **Marital status** | | | |
| | Single | 455 | (43.4%) |
| | Married/Living as couple | 562 | (53.7%) |
| | Widowed/Divorced/Separated | 30 | (2.9%) |
| **Employment status** | | | |
| | Employed for wages | 538 | (51.4%) |
| | Self-employed | 74 | (7.1%) |
| | Out of work/Unemployed | 75 | (7.2%) |
| | A student | 259 | (24.7%) |
| | Retired | 23 | (2.2%) |
| | Unable to work | 9 | (0.85%) |
| | Problem caused by COVID-19 | 59 | (5.6%) |
| | Other | 10 | (0.95%) |
| **Health state** | | | |
| | Healthy | 956 | (91.3%) |
| | With risk factors for cardiovascular disease | 81 | (7.7%) |
| | With cardiovascular disease | 10 | (1%) |

$d$ = 0.44). For most questions, an increased score was noted with the following exceptions: "I was a bad person" and "I did everything wrong". Particularly, bad-feeling related questions such as unhappy, unenjoyed, tired, hated himself, no good and lonely, showed higher score at "during" compared to "before" confinement with |Δ%| ranged from 37% to 107% ($5.07 \leq z \leq 12.60$; $p < .001$, $0.17 \leq d \leq 0.47$). Similarly, scored responses to questions related to how the subject has been acting (i.e., restless, crying and doing nothing) or thinking (i.e., not properly, not concentrated, unloved and not good as others) in bad way showed higher score at "during" compared to "before" confinement with |Δ%| ranged from 10% to 76% ($2.30 \leq z \leq 9.82$; $.45 \leq p \leq .001$, $0.07 \leq d \leq 0.46$). For detailed distribution of responses (in %) please see S2 Table.

**Table 2. Responses to the Short Warwick-Edinburgh Mental Wellbeing Scale before and during home confinement.**

| Questions | Before confinement | During confinement | Δ (Δ%) | 95% IC | *t* test | *p* value | Cohen's *d* |
|---|---|---|---|---|---|---|---|
| 1. I've been feeling optimistic about the future | 4.08±0.91 | 3.54±1.11 | -0.54 (-13.2%) | 0.49–0.59 | 20.260 | < .001 | 0.626 |
| 2. I've been feeling useful | 4.05±0.89 | 3.62±1.13 | -0.43 (-10.7%) | 0.37–0.49 | 14.605 | < .001 | 0.451 |
| 3. I've been feeling relaxed | 3.38±0.94 | 3.25±1.07 | -0.13 (-3.9%) | 0.06–0.21 | 3.442 | < .001 | 0.106 |
| 4. I've been dealing with problems well | 3.88±0.81 | 3.62±0.93 | -0.26 (-6.6%) | 0.21–0.3 | 10.749 | < .001 | 0.332 |
| 5. I've been thinking clearly | 3.99±0.77 | 3.71±0.94 | -0.28 (-6.9%) | 0.22–0.33 | 10.368 | < .001 | 0.320 |
| 6. I've been feeling close to other people | 3.88±0.92 | 3.26±1.16 | -0.61 (-15.8%) | 0.54–0.69 | 16.644 | < .001 | 0.514 |
| 7. I've been able to make up my own mind about things | 4.04±0.83 | 3.72±1.00 | -0.32 (-7.9%) | 0.27–0.37 | 12.887 | < .001 | 0.398 |
| Total score | 27.3±4.37 | 24.73±5.18 | -2.57 (-9.4%) | 2.3–2.84 | 18.821 | < .001 | 0.582 |

## Discussion

The present study reports results from the first 1047 participants who responded to our ECLB-COVID19 multiple languages online survey. Findings indicate significant negative effects of the current COVID-19 pandemic on mental health, especially mental wellbeing, mood, and feeling. There, mental wellbeing (estimate with the total score in SWEMWBS) decreased significantly by 9.4% "during" compared to "before" home confinement with more individuals (+12.89%) reporting a very low to low mental wellbeing. The largest effects of the current COVID-19 pandemic were observed in questions related to optimistic feeling, closed to others, useful, and thinking. Furthermore, results from the mood and feelings questionnaire showed significant increase by 44.9% in SMFQ total score, indicating negative effects with more people (+10%) showing depressive symptoms at "during" compared to "before" home confinement. Especially, questions related to unhappiness, unenjoyment, bad feeling, unclear thinking and loneliness showed highest effect sizes.

The present findings support previous reports suggesting several psychological perturbations and mood disturbances such as stress, depression, irritability, insomnia, fear, confusion, anger, frustration, boredom, and stigma during quarantine periods of earlier infection [14, 30, 31]. Regarding the COVID-19 related research, first results from Chinese studies indicate that the COVID-19 outbreak engendered anxiety, depression, sleep problems, and other psychological problems [32, 33]. The significantly lower total SWEMWBS score and higher total

**Table 3. Responses to the Short Mood and Feelings Questionnaire before and during home confinement.**

| Questions | Before confinement | During confinement | Δ (Δ%) | *z* values | 95% IC | *p* value | Cohen's *d* |
|---|---|---|---|---|---|---|---|
| 1. I felt miserable or unhappy | 0.49±0.57 | 0.79±0.72 | 0.30 (61.2%) | z = 12.124 | -0.34–0.26 | < .001 | 0.458 |
| 2. I didn't enjoy anything at all | 0.29±0.51 | 0.6±0.7 | 0.31 (107.7%) | z = 12.609 | -0.35–0.27 | < .001 | 0.468 |
| 3. I felt so tired I just sat around and did nothing | 0.46±0.6 | 0.81±0.78 | 0.35 (76.2%) | z = 12.456 | -0.39–0.3 | < .001 | 0.460 |
| 4. I was very restless | 0.46±0.6 | 0.66±0.75 | 0.20 (44%) | z = 7.762 | -0.25–0.16 | < .001 | 0.271 |
| 5. I felt I was no good anymore | 0.34±0.53 | 0.55±0.71 | 0.21 (62.3%) | z = 9.822 | -0.25–0.18 | < .001 | 0.351 |
| 6. I cried a lot | 0.39±0.6 | 0.43±0.67 | 0.04 (10.1%) | z = 1.997 | -0.07–0.01 | 0.045 | 0.071 |
| 7. I found it hard to think properly or concentrate | 0.53±0.58 | 0.77±0.74 | 0.24 (45.1%) | z = 9.370 | -0.28–0.20 | < .001 | 0.336 |
| 8. I hated myself | 0.23±0.49 | 0.32±0.6 | 0.09 (37.3%) | z = 5.074 | -0.12–0.06 | < .001 | 0.175 |
| 9. I was a bad person | 0.15±0.39 | 0.17±0.44 | 0.01 (8.6%) | z = 1.121 | -0.04–0.01 | 0.262 | 0.037 |
| 10. I felt lonely | 0.39±0.58 | 0.59±0.73 | 0.2 (52.2%) | z = 8.740 | -0.24 - -0.16 | < .001 | 0.308 |
| 11. I thought nobody really loved me | 0.26±0.52 | 0.29±0.57 | 0.03 (10.2%) | z = 2.296 | -0.05–0.01 | 0.021 | 0.080 |
| 12. I thought I could never be as good as other people | 0.23±0.49 | 0.26±0.54 | 0.04 (16.4%) | z = 3.152 | -0.06–0.02 | < .001 | 0.108 |
| 13. I did everything wrong | 0.27±0.49 | 0.27±0.49 | 0.0 (0.3%) | z = 0.080 | -0.02–0.02 | 0,936 | 0.002 |
| Total score | 4.49±4.41 | 6.5±5.63 | 2.01 (44.9%) | z = 14.520 | -2.29 - -1.73 | < .001 | 0.436 |

SMFQ score "during" compared to "before" confinement, observed in a sample of more than one thousand participants from Western Asian, North Africa and Europe, support the negative effects of the current COVID-19 pandemic on mental wellbeing and emotional state. Taken together, findings from China and from our survey provide insight into the risk of worldwide emotional distress and mental functioning (e.g., low wellbeing, anxiety, depression) during the COVID-19 home confinement period.

Weakening of physical and social contacts with the disruption of normal lifestyles (e.g., lower freedoms, financial losses, sedentariness, sleep disorder, unhealthy diet) during the COVID-19 outbreaks, have been suggested as major risk factors for lower emotional wellbeing and mental disorders [8, 34]. Furthermore, research indicates that some groups may be more vulnerable to the psychosocial effects of the COVID-19 pandemic. Particularly, people with risk factors for COVID-19 infection (e.g., diabetes, chronic heart failure, COPD, immune deficiency), people living in congregate settings (e.g., Hospice) and people with a predisposition and/or pre-existing psychiatric or substance use problems are at increased risk for mental health problems [5].

Since mental disorders have been previously identified as risk factors for several chronic diseases (e.g. hypertension; obesity, dementia) [11, 35–37] and showed to be associated with increased mortality [38, 39] a crisis-oriented interdisciplinary intervention approach to promote wellbeing and mitigate the negative effects of the COVID-19 pandemic on mental health is urgently needed [6, 40–42].

An active lifestyle, including physical and social activity, is an important modifiable factor for mental health across the lifespan [43]. Taking into-consideration that psychosocial tolls of the COVID-19 pandemic appears to be significantly associated with unhealthy lifestyle behaviours including physical and social inactivity, poorer sleep quality as well as unhealthy diet [19, 44], it seems important that this intervention should focus on fostering social communication, physical activity, sleep quality and healthy dietary behaviours [6, 7, 14, 17, 45]. This multidisciplinary intervention can be supported and delivered to the general populations through technology-based solutions such as fitness and nutritional apps, sleep monitoring device, video streaming, exergames, social network, gamification and/or virtual coach.

Furthermore, considering the more vulnerable population to the psychosocial strain, supportive intervention should include "need-oriented" psychosocial services (e.g., psychoeducation, cognitive behavioural techniques, and/or consulting with specialists) delivered by means of telemedicine.

However, to ensure a sustainable intervention approach, future research should investigate the long-term impact of the COVID-19 pandemic on mental health and identify which component(s) of psychosocial strain may persist after the quarantine.

## Strengths, limitations and perspective

The strength of this study is that the data was collected very quickly during the restrictions using a fully anonymous cross-disciplinary survey provided in multiple language and widely distributed in several continents. However, most participants (90.2%) were 55 years old or younger, healthy (90.5%), and educated with a degree beyond high school (90.9%). These demographic characteristics may influence the results, thus the present findings need to be interpreted with caution. Additionally, as cultural differences were previously suggested as relevant factor in moods [46], further large studies analysing differences between countries are warranted. The ECLB-COVID19 survey has since been further translated to Dutch, Persian, Italian, Russian, Indian, Malayalam and Greek languages which has allowed for the addition of more participants and countries. The data will be used in our future post-hoc studies to assess the interaction between the mental and emotional strain evoked by COVID-19 and the

demographical and cultural characteristics of the participants. Identifying exact behavioural changes in each country will be also performed to provide better-informed decisions during pandemics' re-opening process. Regarding the methodological issues, possible limitations could be related to the (i) use of the cross-sectional design assessing the "before" home confinement condition retrospectively and to the (ii) disuse of cookie-based or IP-based duplicate protection to exclude duplicates. However, it should be noted that our consortium elected to avoid IP or cookie safety measures as we know that during home confinement more than one family member can use the same computer (e.g., same IP). Moreover, given that home confinement was a sudden measure in most countries, we were not able to develop and spread the survey at "before" home confinement.

## Conclusion

Besides stresses inherent in the illness itself, results from the ECLB-COVID19 survey reveal a negative effect of home-confinement on mental and emotional wellbeing with more people developing depressive symptoms "during" compared to "before" the confinement period. This increased psychosocial strain triggered by the enforced home confinement should encourage stakeholders and policy makers to implement a crisis-oriented interdisciplinary intervention to mitigate the negative effects of restrictions and to foster an Active and Healthy Confinement Lifestyle (AHCL).

## Supporting information

**S1 File. A copy of the complete ECLB-COVID19 survey's questionnaires.**
(PDF)

**S1 Table. Distribution of responses (%) in each item of the mental wellbeing questionnaire.**
(PDF)

**S2 Table. Distribution of responses (%) in each item of the mood and feeling questionnaire.**
(PDF)

## Acknowledgments

We thank our ECLB-COVID19 consortium's colleagues who provided insight and expertise that greatly assisted the research. The ECLB-COVID19 consortium is leaded by Dr. Achraf Ammar (ammar.achraf@ymail.com) and is composed by the individual authors of the present paper (Full names and affiliations can be found in the authors information section). We thank all other colleagues and peoples who believed on this initiative and helped to distribute the anonymous survey worldwide. We are also immensely grateful to all participants who #StayHome & #BoostResearch by voluntarily taken the #ECLB-COVID19 survey. We would like to acknowledge the recent addition to our team of Dr. Bill McIlroy and Dr. Donald Cowan, both from the University of Waterloo in Canada, who will be participating in the development of information technologies that will support our technology-driven solutions to alleviate some of the serious effects of the COVID-19 pandemic and subsequent quarantine. This manuscript has been released as a pre-print at https://www.medrxiv.org/content/10.1101/2020.05.05.20091058v1, [18].

**Transparency declaration**

The lead author/manuscript's guarantor (Achraf Ammar) affirms that the manuscript is an honest, accurate, and transparent account of the study being reported; that no important aspects of the study have been omitted; and that any discrepancies from the study as planned have been explained.

## Author Contributions

**Conceptualization:** Achraf Ammar, Hamdi Chtourou.

**Data curation:** Achraf Ammar, Khaled Trabelsi, Hamdi Chtourou, Omar Boukhris, Liwa Masmoudi, Bassem Bouaziz, Boštjan Šimunič, Rado Pišot.

**Formal analysis:** Achraf Ammar, Khaled Trabelsi, Hamdi Chtourou, Omar Boukhris, Liwa Masmoudi, Bassem Bouaziz.

**Investigation:** Achraf Ammar, Patrick Mueller, Khaled Trabelsi, Hamdi Chtourou, Omar Boukhris, Liwa Masmoudi, Michael Brach, Marlen Schmicker, Ellen Bentlage, Daniella How, Mona Ahmed, Asma Aloui, Omar Hammouda, Laisa Liane Paineiras-Domingos, Annemarie Braakman-jansen, Christian Wrede, Sophia Bastoni, Carlos Soares Pernambuco, Leonardo Jose Mataruna-Dos-Santos, Morteza Taheri, Khadijeh Irandoust, Aïmen Khacharem, Nicola L. Bragazzi, Jad Adrian Washif, Jordan M. Glenn, Nicholas T. Bott, Faiez Gargouri, Lotfi Chaari, Hadj Batatia, Samira C. khoshnami, Evangelia Samara, Vasiliki Zisi, Parasanth Sankar, Waseem N. Ahmed, Gamal Mohamed Ali, Osama Abdelkarim, Mohamed Jarraya, Kais El Abed, Mohamed Romdhani, Nizar Souissi, Lisette Van Gemert-Pijnen, Stephen J. Bailey, Wassim Moalla, Jonathan Gómez-Raja, Monique Epstein, Robbert Sanderman, Sebastian Schulz, Achim Jerg, Ramzi Al-Horani, Taysir Mansi, Mohamed Jmail, Fernando Barbosa, Fernando Ferreira-Santos, Boštjan Šimunič, Rado Pišot, Andrea Gaggioli, Piotr Zmijewski, Jürgen M. Steinacker, Jana Strahler, Bryan L. Riemann, Notger Mueller, Karim Chamari, Tarak Driss.

**Methodology:** Achraf Ammar, Khaled Trabelsi, Hamdi Chtourou, Omar Boukhris, Michael Brach.

**Project administration:** Achraf Ammar.

**Supervision:** Achraf Ammar, Karim Chamari, Tarak Driss, Anita Hoekelmann.

**Validation:** Achraf Ammar.

**Writing – original draft:** Achraf Ammar, Patrick Mueller.

**Writing – review & editing:** Khaled Trabelsi, Hamdi Chtourou, Omar Boukhris, Michael Brach, Ellen Bentlage, Daniella How, Asma Aloui, Jordan M. Glenn, Nicholas T. Bott, Jana Strahler, Laurel Riemann, Bryan L. Riemann, Notger Mueller, Karim Chamari, Tarak Driss, Anita Hoekelmann.

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
