## [Decision Letter · Decision Letter 0]

13 Jul 2020

PONE-D-20-15158

Emotional consequences of COVID-19 home confinement: The ECLB-COVID19 multicenter study

PLOS ONE

Dear Dr. Ammar,

Thank you for submitting your manuscript to PLOS ONE. After careful consideration, we feel that it has merit but does not fully meet PLOS ONE’s publication criteria as it currently stands. Therefore, we invite you to submit a revised version of the manuscript that addresses the points raised during the review process.

I feel you may improve the manuscript following the comments by two reviewers.

We look forward to receiving your revised manuscript.

Kind regards,

Juan-Carlos Pérez-González, Ph.D.

Academic Editor

PLOS ONE

Journal Requirements:

'The author(s) received no specific funding for this work.'

We note that one or more of the authors are employed by a commercial company: PharmIAD, Inc,Savannah

3. One of the noted authors is a consortium; ECLB-COVID19 Consortium.

In addition to naming the author group, please list the individual authors and affiliations within this group in the acknowledgments section of your manuscript.

Please also indicate clearly a lead author for this group along with a contact email address.

Reviewers' comments:

Reviewer's Responses to Questions

**Comments to the Author**

1. Is the manuscript technically sound, and do the data support the conclusions?

Reviewer #1: Yes

Reviewer #2: Yes

2. Has the statistical analysis been performed appropriately and rigorously? 

Reviewer #1: Yes

Reviewer #2: Yes

3. Have the authors made all data underlying the findings in their manuscript fully available?

Reviewer #1: No

Reviewer #2: No

4. Is the manuscript presented in an intelligible fashion and written in standard English?

Reviewer #1: Yes

Reviewer #2: Yes

5. Review Comments to the Author

Reviewer #1: This is and interesting manuscript and the study topic is very relevant in the current situation. In general, the article is well written and easy to read. The sample number is enough. However, some aspects must be detailed to improve the quality of the article.

GENERAL COMMENTS

The written form of some results could be improved. For example: Statistical symbols letters Could be written in cursive letters or p and r values should be written without “0” before the dot (p = .011). But this depends on the journal. Please check if the results are written correctly according to the journal's criteria

Reading the paper, I understand that the two questionnaires (Short Warwick-Edinburgh Mental Well-being Scale and Short Mood and Feelings Questionnaire) are added to the demographic questions in Table 1. In my opinion, to help other researchers to replicate the study, the complete survey could be attached as a supplementary file. Moreover, this paragraph “The survey included sixty-four questions on health, mental wellbeing, mood, life satisfaction and multidimension lifestyle behaviors (physical activity, diet, social participation, sleep, technology-use, need of psychosocial support).” Suggest that the survey is longer and that only a part of the questionnaire was used in this study.

Regarding the questionnaire, it was translated to “English, German, French, Arabic, Spanish, Portuguese, and Slovenian languages”. This in very interesting as the authors can reach more people and therefore expand the sample. However, the validation process of the different versions has not been explained. On the other hand, the cronbach's alpha values have not been reported. What was the reliability of the questionnaire in this sample?

Results are showed for the total of the sample. However, the survey was sent to different countries and continents which has different isolation conditions when the questionnaire was filled. How might this have affected the results? Would this detail be a possible limitation of the study? Furthermore, cultural differences can be a relevant factor in moods [1].

Another important aspect is the sample distribution which is well balanced in gender but interestingly high educated (Master/doctorate degree 527 (50.3%)). This could generate some bias. Fortunately, the authors have mentioned this in the limitations of the study. Nonetheless, differences by gender has not been reported. In my opinion, it is important to report the existence or not of these differences since the analysis has been carried out men and women together.

MINOR POINTS

Some cites should be revised in the manuscript. For example: “Google’s privacy policy (https://policies.google.com/privacy?hl=en)” or “depression in the respondent.18”.

Authors have use different social networks, although this method has been validated previously [2], how the authors think that this percentage could be affected the sample?

The authors have written the following: “we considered that a score between 7 and13 reflects very low positive mental wellbeing, 14-20 reflects low positive mental wellbeing, 21-27 reflects medium positive mental wellbeing; and 28-35 reflects high positive mental wellbeing.” Why this values or scale? Is there any reference supporting these cut points?

The following discussion paragraph is not supported by the results showed in this study. “The significantly lower total SWEMWBS score and higher total SMFQ score “during” compared to “before” confinement support the negative effects of the current COVID-19 pandemic on mental wellbeing and emotional state in participants from Western Asian, North Africa and Europe.”

Authors interestingly suggest possible solutions to improve health during confinement related to lifestyle and physical activity :“Given that an active lifestyle including physical and social activity is an important modifiable factor for mental health across the lifespan (Rohrer et al. 2005), this intervention should focus on fostering social communication and physical activity (Ammar et al.2020a-c). More references regarding this topic could be added.[3,4]

1. Palinkas, L.A.; Johnson, J.C.; Boster, J.S.; Rakusa-Suszczewski, S.; Klopov, V.P.; Fu, X.Q.; Sachdeva, U. Cross-cultural differences in psychosocial adaptation to isolated and confined environments. Aviation, space, and environmental medicine 2004, 75, 973-980.

2. Browne, K. Snowball sampling: using social networks to research non‐heterosexual women. Int J Soc Res Methodol 2005, 8, 47-60.

3. Xiang, M.; Zhang, Z.; Kuwahara, K. Impact of COVID-19 pandemic on children and adolescents' lifestyle behavior larger than expected. Progress in Cardiovascular Diseases 2020.

4. Brooks, S.K.; Webster, R.K.; Smith, L.E.; Woodland, L.; Wessely, S.; Greenberg, N.; Rubin, G.J. The psychological impact of quarantine and how to reduce it: rapid review of the evidence. The Lancet 2020.

Reviewer #2: This work is a quality study.

Its main contribution is that it focuses on comparing levels of well-being and distress before and during COVID-19 crisis, which is a novel approach to the study of the phenomenon, since studies have usually focused on how mental health is at the time of assessment during confinement or the health emergency.

It also has another advantage, the participants belong to different continents, with a prominent participation of North Africa. As we know most studies provide data from Asia, Europe, and United States or similar, so this is an advantage as well.

Major issues:

In the description of the SWEMWBS cut-off points, it is indicated that the following points were followed in this study: "In this study, we considered that a

score between 7 and13 reflects very low positive mental wellbeing, 14-20 reflects low positive mental wellbeing, 21-27 reflects medium positive mental wellbeing; and 28-35 reflects high positive mental wellbeing." Unlike the Short Mood and Feelings Questionnaire (SMFQ), where it is stated what the use of the cut-off point is based on, this is not the case. It would be necessary to provide the authors' basis for this classification.

Although the decrease in well-being and distress before and after the crisis is established, later in the discussion and conclusions it is recommended to apply Active

and Healthy Confinement Lifestyle (AHCL), a crisis-oriented interdisciplinary

intervention focused on Weakening of physical and social contacts with the disruption of normal lifestyles. From my point of view this suggestion is not well argued on the basis of the current study. The current study establishes that there is a worsening of mental health and well-being in the world population due to COVID-19, but it does not deepen the knowledge of the factors that explain this worsening, so I see it as very pretentious to recommend a specific intervention in this sense. I would like the authors to review this point and to go deeper into the justification of this issue.

Minor issues:

- On page 16 when describing the effect sizes there is a misprint in “Cohn, 1988.”

"Effect size (Cohen's d) was calculated to determine the magnitude of

the change of the score and was interpreted using the following criteria: 0.2 (small), 0.5 (moderate), and 0.8 (large) (Cohn, 1988). Statistical significance was accepted as α<0.05."

- On page 18, there is a room left in "significantly by 9.4 % during home"

6. PLOS authors have the option to publish the peer review history of their article (what does this mean?). If published, this will include your full peer review and any attached files.

Reviewer #1: No

Reviewer #2: No

---

## [Author Response · Author response to Decision Letter 0]

20 Jul 2020

Reviewer #1: 

This is an interesting manuscript and the study topic is very relevant in the current situation. In general, the article is well written and easy to read. The sample number is enough. However, some aspects must be detailed to improve the quality of the article.

The authors would like to thank the reviewer for the insightful and constructive comments on our work. We have carefully considered all of the suggestions and have revised the manuscript accordingly. We believe that our manuscript is much stronger as a result of these modifications.

Please find the authors’ responses to the individual comments below.

GENERAL COMMENTS

The written form of some results could be improved. For example: Statistical symbols letters Could be written in cursive letters or p and r values should be written without “0” before the dot (p = .011). But this depends on the journal. Please check if the results are written correctly according to the journal's criteria

Thank you for your comment. We revised the results section according to your suggestion and with respect to the journal guidelines.

Reading the paper, I understand that the two questionnaires (Short Warwick-Edinburgh Mental Well-being Scale and Short Mood and Feelings Questionnaire) are added to the demographic questions in Table 1. In my opinion, to help other researchers to replicate the study, the complete survey could be attached as a supplementary file. 

Thank you for your suggestion. The complete survey (google form copy) has been attached as a supplementary file (S1 Google form survey). This has been also indicated in the revised text. 

Moreover, this paragraph “The survey included sixty-four questions on health, mental wellbeing, mood, life satisfaction and multidimension lifestyle behaviors (physical activity, diet, social participation, sleep, technology-use, need of psychosocial support).” Suggest that the survey is longer and that only a part of the questionnaire was used in this study.

Yes, the survey including questionnaires related to multiple lifestyle variables (more details on these questionnaires were included in the revised version section: “Survey questionnaires”) and in the present manuscript only data from SWEMWBS and SMFQ were presented. This has been highlighted in the revised version (at the end of the introduction section).

Regarding the questionnaire, it was translated to “English, German, French, Arabic, Spanish, Portuguese, and Slovenian languages”. This in very interesting as the authors can reach more people and therefore expand the sample. However, the validation process of the different versions has not been explained. On the other hand, the cronbach's alpha values have not been reported. What was the reliability of the questionnaire in this sample?

Thank you for your comment. More details about the validation, translation and reliability of the whole survey were added in the revised version as following. 

“The ECLB-COVID19 is a translational electronic survey designed to assess emotional and behavioral change associated with home confinement during the COVID-19 outbreak. Therefore, a collection of validated and/or crisis-oriented brief questionnaires were included (Ammar et al. 2020a-e). These questionnaires assess mental wellbeing (Short Warwick-Edinburgh Mental Wellbeing Scale (SWEMWBS)) [18-20], mood and feeling (Short Mood and Feelings Questionnaire (SMFQ)) [18,19,21], life satisfaction (Short Life Satisfaction Questionnaire for Lockdowns (SLSQL)) [17,19], social participation (Short Social Participation Questionnaire for Lockdowns (SSPQL) [17,19), physical activity (International Physical Activity Questionnaire Short Form (IPAQ-SF)) [6,7,19,22], diet behaviours (Short Diet behaviours Questionnaire for Lockdowns (SDBQL)) [6,7,19], sleep quality (Pittsburgh Sleep Quality Index (PSQI)) [23], and some key questions assessing the technology-use behaviours (Short Technology-use Behaviours Questionnaire for Lockdowns (STBQL)), demographic information, and the need of psychosocial support [19]. Reliability of the shortened and/or newly adopted questionnaires was tested by the project steering group through piloting, prior to survey administration. These brief crisis-oriented questionnaires demonstrated high to excellent test-retest reliability coefficients (r = 0.84-0.96). A multi-language validated version already existed for the majority of these questionnaires and/or questions. However, for questionnaires that did not already exist in multi-language versions, we followed the procedure of translation and back-translation, with an additional review for all language versions from the international scientists of our consortium. In this manuscript, we report only results on mental wellbeing (SWEMWBS), mood, and feeling (SMFQ). A copy of the complete survey can be found in S-1 Google form survey (supplementary file)”

Results are showed for the total of the sample. However, the survey was sent to different countries and continents which has different isolation conditions when the questionnaire was filled. How might this have affected the results? Would this detail be a possible limitation of the study? Furthermore, cultural differences can be a relevant factor in moods [1].

Thank you for your comment. We agree that cultural differences can be an important moderator of the emotional consequences of home confinement. We have highlighted this limitation in the revised manuscript (section “strengths, limitations and perspectives”). We are already working on identifying all possible moderators using the entire dataset in our future manuscripts.

Indeed, the present paper is a part of the whole ECLB-COVID19 project, in which the first step is to understand and to confirm the psychosocial strain of COVID-19 home confinement and the behavioral changes in the general population. The next step will be to identify possible moderators such as demographical, cultural, and/or geographic variables, as well as the restrictions adopted by the included countries. To confirm the presence of psychosocial strain in the general population, data from the first thousand responders were used in this paper. However, our consortium plan to identify the aforementioned moderators using the final collected data (6000-10000 responses). In the second step, a between group (e.g., countries, age group, gender, educational level etc.) analysis will be performed. Using the entire dataset will allow our consortium to perform a between country comparison when home confinement measures end in all countries. For example, comparison between more and low affected countries OR countries with more and lower restrictions will give more insight into the emotional consequence of this pandemics and possible reopening measures for each country.

Regarding the preliminary data presented in this manuscript, we believe that the global community, especially countries which start the re-opening measures (e.g., Germany, Tunisia, Italy, Spain) OR are still imposing total or partial home-confinement (e.g., Iran), are in need of these preliminary results to help understand the emotional consequences of the covid-19 pandemic. Identifying specific psychological changes will allow for better-informed decisions during the re-opening process. 

These points have been highlighted in the limitation and perspectives section of the revised manuscript as following:

“However, given that most participants (90.2%) were 55 years old or younger, healthy (90.5%), and educated with a degree beyond high school (90.9%). These demographic characteristics may influence the results, thus the present findings need to be interpreted with caution. Additionally, as cultural differences were previously suggested be a relevant factor in moods [46], further large studies analyzing differences between countries are warranted. The ECLB-COVID19 survey has been also translated to Dutch, Persian, Italian, Russian, Indian, Malayalam and Greek languages which has allowed for the addition of more participants and countries. The data will be used in our future post-hoc studies to assess the interaction between the mental and emotional strain evoked by COVID-19 and the demographical and cultural characteristics of the participants. Identifying exact behavioural changes in each country will be also performed to provide better-informed decisions during pandemics’ re-opening process”

Another important aspect is the sample distribution which is well balanced in gender but interestingly high educated (Master/doctorate degree 527 (50.3%)). This could generate some bias. Fortunately, the authors have mentioned this in the limitations of the study. Nonetheless, differences by gender has not been reported. In my opinion, it is important to report the existence or not of these differences since the analysis has been carried out men and women together.

Thank you for your comments. This point has been highlighted in the limitation section and as we mentioned in the previous responses, analyzing differences by demographical and cultural characteristics with other possible moderators using the entire dataset are our future goals within the ECLB-COVID19 project.

MINOR POINTS

Some cites should be revised in the manuscript. For example: “Google’s privacy policy (https://policies.google.com/privacy?hl=en)” or “depression in the respondent.18”.

Thank you for your comment. We corrected these errors.

Authors have use different social networks, although this method has been validated previously [2], how the authors think that this percentage could be affected the sample?

In such crisis with physical and social distancing, using electronic survey was the only safe way to collect data and understand the psychosocial effect of pandemics. The consortium is aware that through using electronic survey, some no-accurate responses can be collected, and can bias the results. To reduce this bias, our consortium tried to collect as many responses as possible during a short period through approaching participants via official email-invitation and institute website, but also via different social media platforms. Additionally, in the consent participation, participants were requested to be honest in their response. By collecting, 1000 responses during the first week and up to 5000 responses during the first month, this strategy demonstrated high efficiency. 

The authors have written the following: “we considered that a score between 7 and13 reflects very low positive mental wellbeing, 14-20 reflects low positive mental wellbeing, 21-27 reflects medium positive mental wellbeing; and 28-35 reflects high positive mental wellbeing.” Why this values or scale? Is there any reference supporting these cut points?

Thank you for your comment. We adjusted the cut-off points in the revised version according to Stranges et al. 2014 and Ng Fat et al. 2017. Indeed, based on scores that were at least one standard deviation below and above the mean, respectively (Stranges et al. 2014), categories for SWEMWBS were considered ‘low’ (7–19.3), ‘medium’ (20.0–27.0) and ‘high’ (28.1–35) mental wellbeing (Ng Fat et al. 2017). 

The following sentences were added in the subsection: The Short Warwick-Edinburgh Mental Well-being Scale (SWEMWBS) “Total scores range from 7 to 35 with higher scores indicating higher positive mental wellbeing. Based on scores that were at least one standard deviation below and above the mean, respectively [26], categories for SWEMWBS were considered ‘low’ (7–19.3), ‘medium’ (20.0–27.0) and ‘high’ (28.1–35) mental wellbeing [20].”

The following discussion paragraph is not supported by the results showed in this study. “The significantly lower total SWEMWBS score and higher total SMFQ score “during” compared to “before” confinement support the negative effects of the current COVID-19 pandemic on mental wellbeing and emotional state in participants from Western Asian, North Africa and Europe.”

Thank you for your comment. 

The first one thousand responders to our survey are from Western Asian, North Africa and Europe. The SWEMWBS total score decreased significantly by 9.4% during compared to before home confinement, the SMFQ total score increased significantly by 44.9% “during” compared to “before” home confinement. Lower SWEMWBS was previously linked to low mental wellbeing (Ng Fat et al. 2017), while higher SMFQ was previously suggested to indicate the presence of depression in the respondent (Thabrew et al. 2018). Therefore, we indicated that results support the negative effects of the current COVID-19 pandemic on mental wellbeing and emotional state in the present survey participants. 

However, as we did not analyses the data of western Asia, North Africa and Europe separately we reformulated this paragraph, as following, to avoid any misunderstanding: “The significantly lower total SWEMWBS score and higher total SMFQ score “during” compared to “before” confinement, observed in a sample of one thousand participants from Western Asian, North Africa and Europe, support the negative effects of the current COVID-19 pandemic on mental wellbeing and emotional state”.

Authors interestingly suggest possible solutions to improve health during confinement related to lifestyle and physical activity :“Given that an active lifestyle including physical and social activity is an important modifiable factor for mental health across the lifespan (Rohrer et al. 2005), this intervention should focus on fostering social communication and physical activity (Ammar et al.2020a-c). More references regarding this topic could be added3,4]

Thank you for the suggested references. Both references were added in the revised version

1. Palinkas, L.A.; Johnson, J.C.; Boster, J.S.; Rakusa-Suszczewski, S.; Klopov, V.P.; Fu, X.Q.; Sachdeva, U. Cross-cultural differences in psychosocial adaptation to isolated and confined environments. Aviation, space, and environmental medicine 2004, 75, 973-980.

2. Browne, K. Snowball sampling: using social networks to research non‐heterosexual women. Int J Soc Res Methodol 2005, 8, 47-60.

3. Xiang, M.; Zhang, Z.; Kuwahara, K. Impact of COVID-19 pandemic on children and adolescents' lifestyle behavior larger than expected. Progress in Cardiovascular Diseases 2020.

4. Brooks, S.K.; Webster, R.K.; Smith, L.E.; Woodland, L.; Wessely, S.; Greenberg, N.; Rubin, G.J. The psychological impact of quarantine and how to reduce it: rapid review of the evidence. The Lancet 2020.

Reviewer #2: This work is a quality study.

Its main contribution is that it focuses on comparing levels of well-being and distress before and during COVID-19 crisis, which is a novel approach to the study of the phenomenon, since studies have usually focused on how mental health is at the time of assessment during confinement or the health emergency.

It also has another advantage, the participants belong to different continents, with a prominent participation of North Africa. As we know most studies provide data from Asia, Europe, and United States or similar, so this is an advantage as well.

The authors would like to thank the reviewer for the insightful and constructive comments on our work. We have carefully considered all of the suggestions and have revised the manuscript accordingly. We believe that our manuscript is much stronger as a result of making these modifications.

Please find below the authors’ responses to the individual comments

Major issues:

In the description of the SWEMWBS cut-off points, it is indicated that the following points were followed in this study: "In this study, we considered that a

score between 7 and13 reflects very low positive mental wellbeing, 14-20 reflects low positive mental wellbeing, 21-27 reflects medium positive mental wellbeing; and 28-35 reflects high positive mental wellbeing." Unlike the Short Mood and Feelings Questionnaire (SMFQ), where it is stated what the use of the cut-off point is based on, this is not the case. It would be necessary to provide the authors' basis for this classification.

Thank you for your comment. We adjusted the cut-off points in the revised version (subsection: “The Short Warwick-Edinburgh Mental Well-being Scale (SWEMWBS)”) according to Stranges et al. 2014 and Ng Fat et al. 2017. Indeed, based on scores that were at least one standard deviation below and above the mean, respectively (Stranges et al. 2014), categories for SWEMWBS were considered ‘low’ (7–19.3); ‘medium’ (20.0–27.0) and ‘high’ (28.1–35) mental wellbeing (Ng Fat et al. 2017).

Although the decrease in well-being and distress before and after the crisis is established, later in the discussion and conclusions it is recommended to apply Active

and Healthy Confinement Lifestyle (AHCL), a crisis-oriented interdisciplinaryintervention focused on Weakening of physical and social contacts with the disruption of normal lifestyles. From my point of view this suggestion is not well argued on the basis of the current study. The current study establishes that there is a worsening of mental health and well-being in the world population due to COVID-19, but it does not deepen the knowledge of the factors that explain this worsening, so I see it as very pretentious to recommend a specific intervention in this sense. I would like the authors to review this point and to go deeper into the justification of this issue.

Thank you for your comment. This point has been reviewed in the revised discussion section and the following paragraph was added.

“An active lifestyle, including physical and social activity, is an important modifiable factor for mental health across the lifespan (Rohrer et al. 2005). Taking into-consideration that psychosocial tolls of the COVID-19 pandemic appears to be significantly associated with unhealthy lifestyle behaviours including physical and social inactivity, poorer sleep quality as well as unhealthy diet (Ammar et al. 2020e, Xiang et al. 2020), it seems important that this intervention should focus on fostering social communication, physical activity, sleep quality and healthy dietary behaviours (Ammar et al. 2020a-c; Brooks et al. 2020). This multidisciplinary intervention can be supported and delivered to the general populations through technology-based solutions such as fitness and nutritional apps, sleep monitoring device, video streaming, exergames, social network, gamification, and/or virtual coach.”

Minor issues:

- On page 16 when describing the effect sizes there is a misprint in “Cohn, 1988.”

"Effect size (Cohen's d) was calculated to determine the magnitude of the change of the score and was interpreted using the following criteria: 0.2 (small), 0.5 (moderate), and 0.8 (large) (Cohn, 1988). Statistical significance was accepted as α<0.05."

Thank you for your comment. Correction done 

- On page 18, there is a room left in "significantly by 9.4 % during home"

Thank you for your comment. Correction done

---

## [Decision Letter · Decision Letter 1]

23 Sep 2020

Psychological consequences of COVID-19 home confinement: The ECLB-COVID19 multicenter study

PONE-D-20-15158R1

Dear Dr. Ammar,

We’re pleased to inform you that your manuscript has been judged scientifically suitable for publication and will be formally accepted for publication once it meets all outstanding technical requirements.

Kind regards,

Juan-Carlos Pérez-González, Ph.D.

Academic Editor

PLOS ONE

Additional Editor Comments (optional):

Reviewers' comments:

Reviewer's Responses to Questions

**Comments to the Author**

1. If the authors have adequately addressed your comments raised in a previous round of review and you feel that this manuscript is now acceptable for publication, you may indicate that here to bypass the “Comments to the Author” section, enter your conflict of interest statement in the “Confidential to Editor” section, and submit your "Accept" recommendation.

Reviewer #1: All comments have been addressed

Reviewer #2: All comments have been addressed

2. Is the manuscript technically sound, and do the data support the conclusions?

Reviewer #1: Yes

Reviewer #2: Yes

3. Has the statistical analysis been performed appropriately and rigorously? 

Reviewer #1: Yes

Reviewer #2: Yes

4. Have the authors made all data underlying the findings in their manuscript fully available?

Reviewer #1: Yes

Reviewer #2: Yes

5. Is the manuscript presented in an intelligible fashion and written in standard English?

Reviewer #1: Yes

Reviewer #2: Yes

6. Review Comments to the Author

Reviewer #1: In my opinion the authors have make a good job. The paper now is clearer and more replicable. The method section has been improved and the paper it is very interesting. However, I would make small suggestions which could improve the manuscript:

Authors write the following paragraph: “Reliability of the shortened and/or newly adopted questionnaires was tested by the project steering group through piloting, prior to survey administration. These brief crisis-oriented questionnaires demonstrated high to excellent test-retest reliability coefficients (r = 0.84-0.96).”. This paragraph is confused. It seems that the reliability of the questionnaires was analysed in the pilot study and not with the actual data. Why with the pilot study and not with the used data in this manuscript? If this is the case, authors could analyse the reliability of the used questionnaires if possible. Specially, as the sample could have some bias as they have written in the limitation section. Moreover, why they use the “r” instead “α” to show the reliability coefficient?

Regarding the instrument and all the questionnaires used, it seems that the length of the survey was large which could affect the response rate. In addition, authors have not informed about the response rate. In my opinion it would be interesting to add information about the response rate. If this is not possible because the authors have used the snowball sampling technique, maybe they should add some sentence in the limitations section. In any case, I think the sample, or the number of participants can be representative enough. Adding any reference regarding this aspect could make more robust the method section. The following references could help.

Deutskens E, De Ruyter K, Wetzels M, Oosterveld P. Response rate and response quality of internetbased surveys: An experimental study. Mark Lett. 2004; 15(1): 21-36. https://doi.org/10.1023/B:MARK. 0000021968.86465.00).

Mavletova, A.; Couper, M.P. Mobile web survey design: scrolling versus paging, SMS

versus e-mail invitations. Journal of Survey Statistics and Methodology 2014, 2, 498-

518.

Browne, K. (2005). Snowball sampling: using social networks to research non‐heterosexual women. International journal of social research methodology, 8(1), 47-60.

Lastly authors informed that the total number of questions was: “The survey included sixty-four questions on health, mental wellbeing, mood, life satisfaction and multidimension lifestyle behaviors” in the method section, but this does not match with the “Supporting Information S1 Google form survey.pdf” please revise this aspect.

Reviewer #2: The authors have responded all suggestions and comments by reviewers. The manuscript has improved its quality, so I think the manuscript should be published as it is.

7. PLOS authors have the option to publish the peer review history of their article (what does this mean?). If published, this will include your full peer review and any attached files.

Reviewer #1: No

Reviewer #2: No

---

## [Editor Report · Acceptance letter]

22 Oct 2020

PONE-D-20-15158R1 

Psychological consequences of COVID-19 home confinement: The ECLB-COVID19 multicenter study 

Dear Dr. Ammar:

I'm pleased to inform you that your manuscript has been deemed suitable for publication in PLOS ONE. Congratulations! Your manuscript is now with our production department. 

Kind regards, 

on behalf of

Dr. Juan-Carlos Pérez-González 

Academic Editor

PLOS ONE